# PACAP neurons in the ventral premammillary nucleus regulate reproductive function in the female mouse

Rachel A Ross[1,2,3,4]\*, Silvia Leon[3,5], Joseph C Madara[3,6], Danielle Schafer[7], Chrysanthi Fergani[3,5], Caroline A Maguire[5], Anne MJ Verstegen[3,6], Emily Brengle[5,6], Dong Kong[8,9], Allan E Herbison[7], Ursula B Kaiser[3,5], Bradford B Lowell[3,6], Victor M Navarro[3,5]\*

[1]Department of Psychiatry, Beth Israel Deaconess Medical Center, Boston, United States; [2]Department of Psychiatry, Massachusetts General Hospital, Massachusetts, United States; [3]Harvard Medical School, Massachusetts, United States; [4]McLean Hospital, Boston, United States; [5]Department of Medicine, Division of Endocrinology, Diabetes and Hypertension, Brigham and Women's Hospital, Boston, United States; [6]Department of Medicine, Division of Endocrinology, Diabetes and Metabolism, Beth Israel Deaconess Medical Center, Boston, United States; [7]Centre for Neuroendocrinology, Otago School of Medical Sciences, University of Otago, Dunedin, New Zealand; [8]Department of Neuroscience, Tufts University School of Medicine, Massachusetts, United States; [9]Sackler School of Graduate Biomedical Sciences, Tufts University, Boston, United States

\*For correspondence:
rross4@partners.org (RAR);
vnavarro@bwh.harvard.edu (VMN)

Competing interests: The authors declare that no competing interests exist.

**Abstract** Pituitary adenylate cyclase activating polypeptide (PACAP, *Adcyap1*) is a neuromodulator implicated in anxiety, metabolism and reproductive behavior. PACAP global knockout mice have decreased fertility and PACAP modulates LH release. However, its source and role at the hypothalamic level remain unknown. We demonstrate that PACAP-expressing neurons of the ventral premamillary nucleus of the hypothalamus (PMV^PACAP) project to, and make direct contact with, kisspeptin neurons in the arcuate and AVPV/PeN nuclei and a subset of these neurons respond to PACAP exposure. Targeted deletion of PACAP from the PMV through stereotaxic virally mediated cre- injection or genetic cross to LepR-i-cre mice with *Adcyap1*^fl/fl mice led to delayed puberty onset and impaired reproductive function in female, but not male, mice. We propose a new role for PACAP-expressing neurons in the PMV in the relay of nutritional state information to regulate GnRH release by modulating the activity of kisspeptin neurons, thereby regulating reproduction in female mice.
DOI: https://doi.org/10.7554/eLife.35960.001

## Introduction

The hypothalamic-pituitary-gonadal (HPG) axis is tightly regulated throughout development to ensure the proper timing of puberty onset and attainment of fertility. Metabolic cues play a critical role in this regulatory process by modulating the release of kisspeptin and/or gonadotropin-releasing hormone (GnRH) at the hypothalamic level (*Navarro and Kaiser, 2013*). Situations of energy deficit and surfeit are known to cause fertility impairments (e.g. hypothalamic amenorrhea in anorexia nervosa) through the suppression of pulsatile LH secretion, but the mechanism by which metabolic

factors regulate gonadotrophic axis function is largely unknown (*Laughlin et al., 1998*). Kisspeptins, secreted from kisspeptin (Kiss1) neurons in the arcuate (ARC), anteroventral periventricular and periventricular (AVPV/PeN), are the main secretagogues of GnRH. Kiss1 neurons convey most of the regulatory cues of the gonadotropic axis, which determines the proper pulsatile or surge-like pattern of release of GnRH (*Herbison, 2016*). Despite this primary role of kisspeptin and GnRH neurons in regulating the HPG axis, the signal transduction of metabolic information to these neurons is yet to be fully elucidated.

In this context, pituitary adenylate cyclase-activating polypeptide (PACAP, official gene symbol *Adcyap1*) has emerged as a neuroendocrine factor involved in the regulation of food intake and gonadotropin release (*Hawke et al., 2009*; *Resch et al., 2013*; *Tanida et al., 2013*; *Halvorson, 2014*). PACAP has been implicated in central processes such as development of anxiety behavior (*Mustafa, 2013*) and circadian rhythms (*Mertens et al., 2007*) and, importantly, both the ligand and the receptor are present in a number of metabolic and reproductive nuclei in the hypothalamus, including the ventromedial hypothalamus (VMH) (*Vaudry et al., 2000*). The presence of PACAP in these nuclei has been linked to the anorectic action of leptin at the hypothalamic level by modulating the activity of POMC neurons directly (*Hawke et al., 2009*; *Resch et al., 2013*; *Tanida et al., 2013*; *Martin et al., 2014*; *Mounien et al., 2006*), but this has not been linked to its role in fertility. It is known that whole body deletion of PACAP signaling decreases fertility (*Jamen et al., 2000*) and central PACAP administration delays puberty onset in rodents (*Szabó et al., 2002*); however, its action on gonadotropin release remains controversial as both stimulatory and inhibitory actions have been reported in rodents after central PACAP infusion (*Köves et al., 2014*; *Köves et al., 1998*), indicating a degree of complexity that may be species- and/or sex-specific. Interestingly, a number of studies have invoked PACAP in the metabolic role of leptin (*Hawke et al., 2009*; *Tanida et al., 2013*); however, whether this action is direct (PACAP released from leptin receptor (LepR) expressing neurons) or indirect remains to be determined. In the context of reproduction, while LepR is expressed in a subset of kisspeptin neurons (*Smith et al., 2006*), genetic studies have demonstrated that the main site of leptin's action to regulate reproduction is not on kisspeptin neurons directly, but rather through cells in the ventral premammillary nucleus (PMV) (*Cavalcante et al., 2014*; *Donato et al., 2011*; *Donato et al., 2009*; *Cravo et al., 2013*). We observed that PACAP is highly expressed in this nucleus and therefore we sought to investigate if PACAP in the PMV is important in the regulation of reproductive function and whether it serves as a mediator of leptin to exert its reproductive role.

We generated conditional PACAP knockout (*Adcyap1*[fl/fl]) mice that allow for the deletion of the PACAP gene in the presence of cre-recombinase by genetic cross and viral injection, and we investigated whether PACAP is essential for full reproductive capabilities of female mice. We also used PACAP-i-cre mice (*Krashes et al., 2014*) that allowed us to investigate whether PMV[PACAP] neurons synaptically connect to kisspeptin neurons at different neuroanatomical levels (i.e. ARC and AVPV/PeN) to determine how these PMV[PACAP] neurons fit into the central regulation of the reproductive axis.

## Results

### PACAP release from leptin-responsive neurons is essential for normal timing of puberty onset and fertility

An intact PMV is required to transduce metabolic information through leptin signaling to allow for HPG activity (*Donato et al., 2011*). We found that the PMV is the region of the brain with the highest level of co-localization of pStat3 (an indirect marker of LepR activity) and PACAP (*Figure 1a*); approximately 87% of the PACAP neurons also express pStat3 in that region, though only 70% of the LepR neurons co-express PACAP. There are two other regions, both in the hypothalamus, that also show co-localization of pStat3 and PACAP, although to a lesser extent: the central nucleus of the ventromedial hypothalamus and the supramammillary nucleus (*Figure 1—figure supplement 1*). To investigate if PACAP is an important relay for leptin we produced mice with PACAP deleted conditionally from leptin receptor expressing neurons using the leptin receptor cre knock-in mouse (*Leshan et al., 2006*) crossed to the *Adcyap1*[fl/fl] mouse that we made. The LepR-i-cre mouse expresses cre recombinase in cells that produce the long-form of the leptin receptor, which are

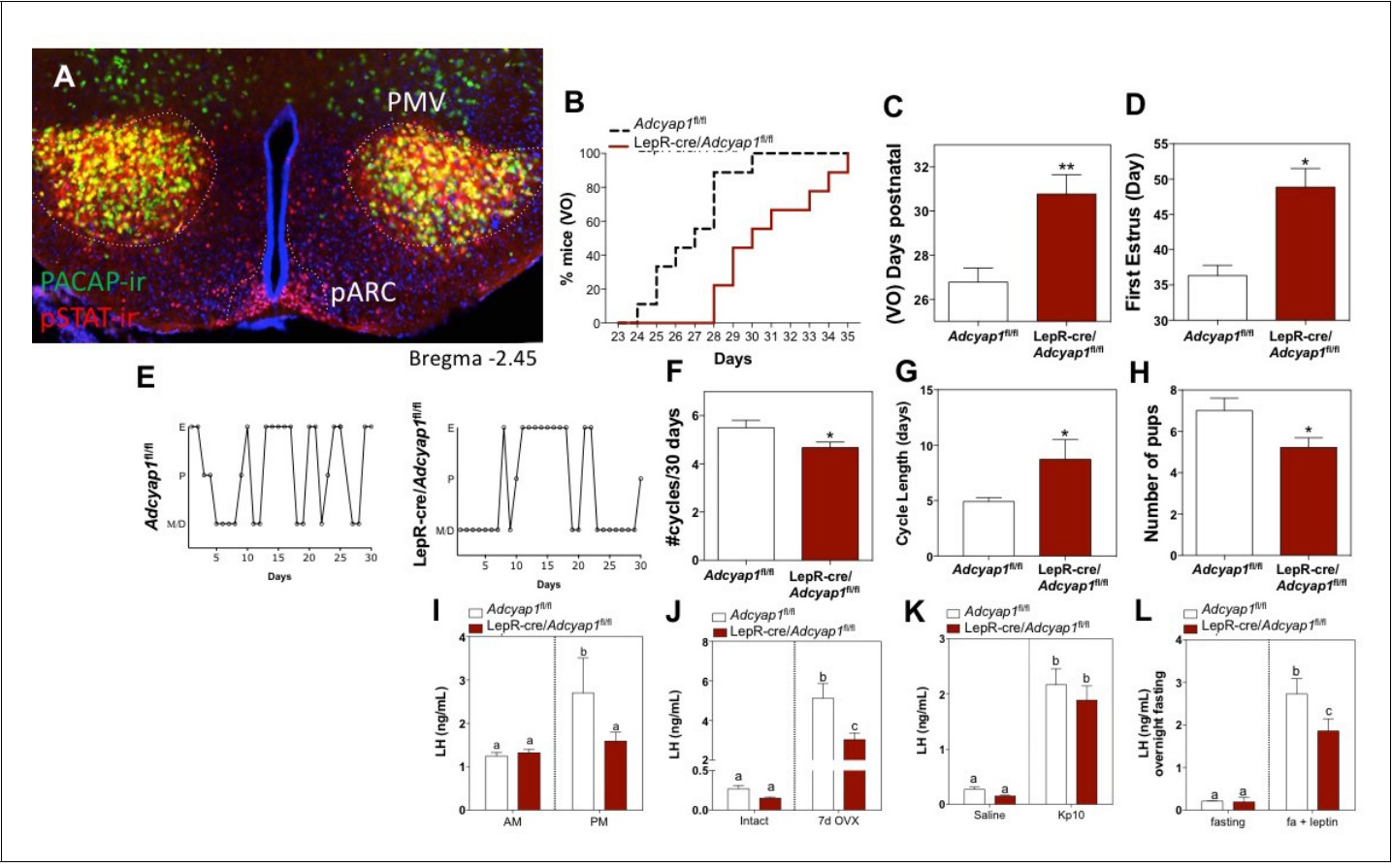

**Figure 1.** PACAP release from leptin-responsive neurons is essential for normal timing of puberty onset and fertility. (A) Representative microphotograph of a coronal section depicting PACAP (green) and pSTAT (red) immunoreactivity in the PMV of adult female PACAP-i-cre;L10-GFP mice treated with 5 mg/kg leptin two hours before perfusion. Yellow cells show co-localization. (B–D) Puberty onset is delayed in LepR- $Adcyap1^{fl/fl}$ mice. (B) Accumulated percentage of mice with vaginal opening (VO), an indirect assessment of puberty onset by daily visualization for two weeks post wean ($Adcyap1^{fl/fl}$ n = 9; LepRcre- $Adcyap1^{fl/fl}$ n = 9). (C) Mean postnatal day of VO ($Adcyap1^{fl/fl}$ n = 9; LepRcre- $Adcyap1^{fl/fl}$ n = 9). **p<0.01, Student t-test. (D) Mean postnatal day of first estrous as determined by histology samples after VO taken over the course of one month. ($Adcyap1^{fl/fl}$ n = 4; LepRcre-$Adcyap1^{fl/fl}$ n = 7). *p<0.05, Student t-test. (E–G) Estrous cyclicity is dysregulated in LepR-$Adcyap1^{fl/fl}$ mice. (E) Representative examples of daily (30 days) estrous cycle phases of control $Adcyap1^{fl/fl}$ (n = 9) and LepRcre- $Adcyap1^{fl/fl}$ mice (n = 9). Mouse estrous cycle is 4 days on average, including estrous phase (E), proestrous phase (P) and met/diestrous phase (M/D) which are combined due to similarity by histology. (F) Mean number of estrous cycles in 30 days. *p<0.05 Student t-test ($Adcyap1^{fl/fl}$ n = 9; LepRcre-$Adcyap1^{fl/fl}$ n = 9). (G) Mean cycle length in days. *p<0.05 Student t-test ($Adcyap1^{fl/fl}$ n = 9; LepRcre-$Adcyap1^{fl/fl}$ n = 9). (H) Number of pups per litter for three sets of breeding pairs over five months of ongoing breeding is significantly lower in LepR- $Adcyap1^{fl/fl}$ mice. *p<0.05 Student t-test ($Adcyap1^{fl/fl}$ n = 18; LepRcre- $Adcyap1^{fl/fl}$ n = 18). (I) Preovulatory LH surge is blunted in LepR-$Adcyap1^{fl/fl}$ mice. Circulating levels of LH in the morning (AM 10:00 hr) and afternoon (PM 19:00 hr) under an LH-surge inducing protocol. $Adcyap1^{fl/fl}$ n = 5; LepRcre- $Adcyap1^{fl/fl}$ n = 5). Different letters indicate statistically different values (2 Way ANOVA followed by Fisher's *post hoc* test, p<0.05). (J) Circulating LH levels before and one week after OVX. The lack of steroid hormone feedback in control animals leads to a marked increase in LH, which is blunted in LepR-$Adcyap1^{fl/fl}$ mice. Different letters indicate statistically different values (2 Way ANOVA followed by Bonferroni *post hoc* test, p<0.05), (intact: $Adcyap1^{fl/fl}$ n = 7; LepRcre-$Adcyap1^{fl/fl}$ n = 4. OVX: $Adcyap1^{fl/fl}$ n = 3; LepRcre-$Adcyap1^{fl/fl}$ n = 5). (K) Circulating LH levels 25 min after the injection of vehicle (saline) or 1 nmol kisspeptin 10 (kp10) are not significantly different between the knockout and control animals. Different letters indicate statistically different values (2 Way ANOVA followed by Bonferroni *post hoc* test, p<0.05), (saline: $Adcyap1^{fl/fl}$ n = 7; LepRcre-$Adcyap1^{fl/fl}$ n = 3. Kp10: $Adcyap1$ n = 4; LepRcre-$Adcyap1^{fl/fl}$ n = 9). (L) Circulating LH levels after overnight fast and 30 min after central leptin administration are blunted in LepR-$Adcyap1^{fl/fl}$ mice. Different letters indicate statistically different values (2 Way ANOVA followed by Fisher's *post hoc* test, p<0.05), (fasting: $Adcyap1^{fl/fl}$ n = 3; LepRcre-$Adcyap1^{fl/fl}$ n = 7. Fasting + leptin: $Adcyap1^{fl/fl}$ n = 3; LepRcre-$Adcyap1^{fl/fl}$ n = 9).

DOI: https://doi.org/10.7554/eLife.35960.002

The following figure supplements are available for figure 1:

**Figure supplement 1.** Representatives images of hypothalamic nuclei coronal sections depicting areas of co-labeling (yellow) of PACAP-ir (green) and pSTAT-ir (red) in adult female PACAP-i-cre;L10-GFP mice treated with 5 mg/kg IP leptin two hours before perfusion.
DOI: https://doi.org/10.7554/eLife.35960.003

**Figure supplement 2.** Validation of PACAP conditional allele.

*Figure 1 continued on next page*

Figure 1 continued

DOI: https://doi.org/10.7554/eLife.35960.004

**Figure supplement 3.** Additional characterization of the metabolic and reproductive phenotype of LepR$^{Cre}$-Adcyap1$^{fl/fl}$ male and female mice.
DOI: https://doi.org/10.7554/eLife.35960.005

found primarily in the brain (*Leshan et al., 2006*), thus producing a conditional knockout of PACAP from neurons that express the leptin receptor. We validated this conditional knockout by qPCR and by RNA in situ hybridization (*Figure 1—figure supplements 2* and *3a*).

Because this genetic recombination occurs before puberty onset, we were able to address the question of the role that PACAP in LepR neurons may play in puberty, which relies on a normal functioning HPG axis. Compared to non-cre expressing littermate controls, LepR-i-cre; *Adcyap1*$^{fl/fl}$ females had delayed onset of puberty, measured both by vaginal opening and day of first estrus (*Figure 1b–d*). There was no significant change in body weight in any of the animals on regular chow or high fat diet (*Figure 1—figure supplement 3b*). As expected, deletion of PACAP from LepR neurons leads to dysregulated estrous cycling (*Figure 1e–g*), though less markedly than the lesion of the PMV region (*Donato et al., 2011*); these LepR-i-cre;*Adcyap1*$^{fl/fl}$ females were able to get pregnant, but had fewer pups per litter (*Figure 1h*). Consistent with the ability to become pregnant, all stages of follicle development were present, including corpora lutea (*Figure 1—figure supplement 3c*).

This conditional genetic KO of PACAP from LepR neurons displayed marked HPG axis dysregulation. The LepR-i-cre;*Adcyap1*$^{fl/fl}$ animals showed a blunted LH surge (*Figure 1i*), which correlates with the dysregulated estrous cycle (*Figure 1e–g*). They also had a blunted response to ovariectomy and fewer pups per litter (*Figure 1h,j*). Since neurons located in the PMV have been described to contact GnRH neurons in addition to kisspeptin neurons (*Donato et al., 2011*), it is possible that GnRH neurons require either PACAP or other PMV signals to elicit a proper response to kisspeptin, which could lead to the reproductive impairment seen in these mice. Therefore, in order to determine if the function of GnRH neurons (and gonadotropes) remains intact in this model, we administered kisspeptin intracerebroventricularly (ICV) to these animals and compared LH release with their corresponding controls. Kisspeptin administration led to a similar induction of LH in both groups (*Figure 1k*). This provides further evidence that GnRH neurons are able to respond normally to kisspeptin and, therefore, the reproductive impairment observed in these mice must rely on, or relay through another neuron that stimulates kisspeptin release. Finally, leptin administration to these animals in the morning after a fast, induced a lower than expected rise in LH release (*Figure 1l*); by comparison to the littermate controls, the LH induction in response to leptin treatment was blunted by 30%. This indicates that PACAP is responsible for transducing some, but not all, of the metabolic information relayed by leptin to the HPG axis.

## PACAP deletion from the PMV in adulthood leads to impaired gonadotrophic function

Because the PMV is the site of greatest co-localization of LepR activity and PACAP, and prior studies detailed the importance of the region to the reproductive action of leptin (*Donato and Elias, 2011*), we investigated the role of PACAP in the PMV directly. To determine if PACAP secreted from the PMV is necessary for HPG axis function, we directly deleted PACAP from the PMV of adult female *Adcyap1*$^{fl/fl}$ mice with bilateral stereotaxic injections of an adenovirus carrying cre-recombinase (*Figure 2a,b*). We again found marked dysregulation of the normal estrous cycle when validated PMV$^{PACAP}$ knockout (KO) mice were compared to littermate controls injected with a control virus (same serotype without cre, *Figure 2c*). Mice lacking PACAP in the PMV had fewer estrous cycles in a 25 day period and resultant longer cycle lengths (*Figure 2d,e*). To note, PMV$^{PACAP}$ KO animals maintained equivalent body weight to control animals throughout the study (*Figure 2—figure supplement 1a*).

In order to determine if this deletion had effects on fertility, we mated the female mice to 8–12 week old male C57Bl/6 wild-type mice for 1 week at a time. Males were removed after this period. Nearly all control mice became pregnant and successfully produced litters within 21 days of mating, indicating successful copulation on the first day of mating to the male. In contrast, the PMV$^{PACAP}$ KO

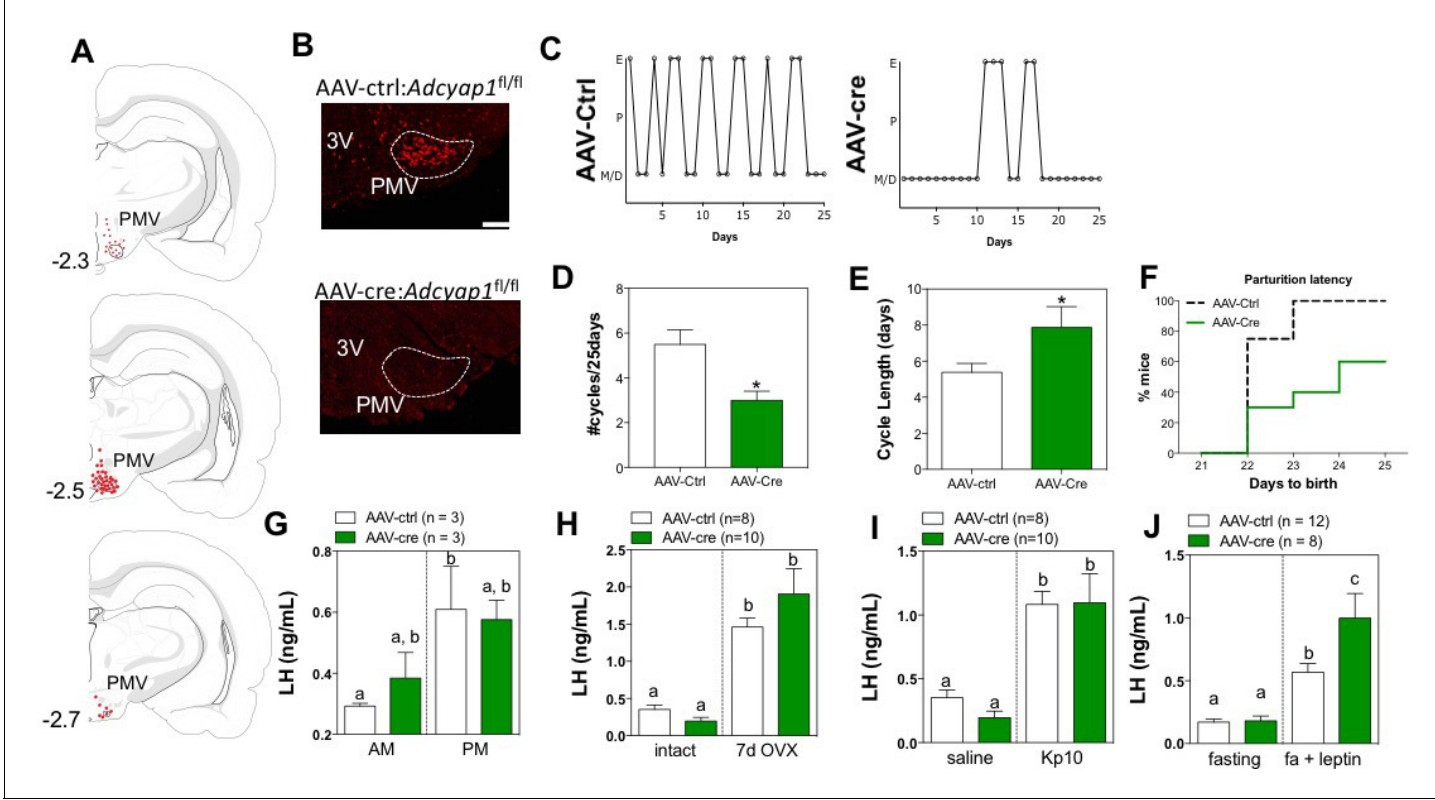

**Figure 2.** Ablation of PACAP expression from the PMV leads to impairment in the gonadotropic axis. (A) Schematic representation of the sites of injection of AAV8.CMV.HI.eGFP-Cre showing the extent of viral spread around the PMV for *Adcyap1*fl/fl study animals. Animals were included in analyses only if they had bilateral injection across >70% of the rostro-caudal diameter of the PMV that did not spread into surrounding areas of PACAP expression (i.e. VMH, dorsal premammillary nucleus). Control virus (AAV8.CMV.PI.eGFP.WPRE.bGH) was injected bilaterally in littermate controls of the same genotype (data not shown). (B) Representative ISH depicting coronal section with *Adcyap1* mRNA in the PMV of AAV8.CMV.PI.eGFP.WPRE.bGH control injected (left panel) and AAV8.CMV.HI.eGFP-Cre injected *Adcyap1*fl/fl female mice. Only animals with bilateral deletion of >70% *Adcyap1* mRNA were included in analyses. (C – E) Assessment of estrous cyclicity shows marked dysregulation in the AAV-cre treated *Adcyap1*fl/fl mice. (C) Representative examples of daily (25 days) estrous cycle phases of control (n = 8) and AAV-cre injected mice (n = 6). (D) Mean number of estrous cycles in 25 days. *p<0.05 Student t-test (AAV-ctrl:*Adcyap1*fl/fl n = 8; AAV-cre:*Adcyap1*fl/fl n = 10). (E) Mean cycle length in days. *p<0.05 Student t-test (control n = 6; AAV-cre n = 8). (F) AAV-cre mediated deletion of PACAP from PMV prolongs parturition latency represented as accumulated percentage of animals giving birth per day after mating with C57Bl/6 male for 1 week (AAV-ctrl:*Adcyap1*fl/fl n = 8; AAV-cre:*Adcyap1*fl/fl n = 6). (G) Analysis of the magnitude of the preovulatory LH surge. Circulating levels of LH in the morning (AM 10:00 hr) and afternoon (PM 19:00 hr) under an LH-surge inducing protocol. (AAV-ctrl:*Adcyap1*fl/fl n = 3; AAV-cre:*Adcyap1*fl/fl n = 3). Different letters indicate statistically different values. Only AAV-Ctrl injected mice showed a significant increase from the AM to the PM. (2 Way ANOVA followed by Bonferroni *post hoc* test, p<0.05). (H) Circulating LH levels before and one week after OVX are not significantly different between the AAV-cre and AAV-ctrl treated *Adcyap1*fl/fl mice. Different letters indicate statistically different values (2 Way ANOVA followed by Bonferroni *post hoc* test, p<0.05), (AAV-ctrl:*Adcyap1*fl/fl n = 8; AAV-cre:*Adcyap1*fl/fl n = 10). (I) Circulating LH levels 25 min after the injection of vehicle (saline) or 1 nmol kisspeptin 10 (kp10) are intact. Different letters indicate statistically different values (2 Way ANOVA followed by Bonferroni *post hoc* test, p<0.05), (AAV-ctrl:*Adcyap1*fl/fl n = 8; AAV-cre:*Adcyap1*fl/fl n = 10). (J) Circulating LH levels after overnight fast and 30 min after central leptin administration indicate PACAP is involved in, but is not necessary for leptin signal transduction through the PMV. Different letters indicate statistically different values (2 Way ANOVA followed by Fisher's *post hoc* test, p<0.05), (AAV-ctrl:*Adcyap1*fl/fl n = 12; AAV-cre: *Adcyap1*fl/fl n = 8).

DOI: https://doi.org/10.7554/eLife.35960.006

The following figure supplement is available for figure 2:

**Figure supplement 1.** PACAP deletion from PMV neurons in adult females does not affect body weight but decreases the number of corpora lutea.
DOI: https://doi.org/10.7554/eLife.35960.007

mice had reduced fecundity, with decreased capacity to produce a litter after mating, and significant delay in the time to become pregnant (*Figure 2f*). Furthermore, those pups that were produced by the PMV^PACAP KO females did not survive more than one day post partum (data not shown), which was thought to be due to neglect/infanticide by the dam based on observation of the animals in

their home cages. Though we could not assess the litter size, we observed a decrease in the number of corpora lutea in these mice (*Figure 2—figure supplement 1b*), which suggests an impairment in ovulation, consistent with a smaller number of follicles maturing per cycle, as a consequence of reduced gonadotropins. In control-injected animals the increase in LH between baseline (10:00) and surge (19:00) is significant, whereas for the KO animals only a trend to increased LH was observed at 19:00, suggesting an alteration in the ability of these mice to mount a normal ovulatory LH surge (*Figure 2g*). On the other hand, the compensatory rise in LH after ovariectomy was not different between the two groups (*Figure 2h*), indicating that the circuitry downstream of gonadal feedback, particularly the ARC[kisspeptin] population, responds normally to the absence of sex steroids.

Similar to what we observed in response to kisspeptin administration in the LepR-i-cre; *Adcyap1*[fl/fl] ablated mice, PMV[PACAP] KO mice had a normal response to kisspeptin administration (*Figure 2i*), indicating intact signaling between kisspeptin and GnRH neurons, and supporting a role for PACAP to modulate kisspeptin release. To consider if PMV[PACAP] is necessary to relay the permissive signal of leptin to gonadotropin release, we administered leptin centrally after an overnight fast to assess whether, as in LepR-i-cre; *Adcyap1*[fl/fl] animals, the magnitude of the LH increase is reduced compared to controls. Unexpectedly, the LH response was not reduced, instead there was a slight but significant increase in LH induction by leptin treatment (*Figure 3j*). Overall these data show that PACAP from the PMV is necessary for normal reproductive function but not for the relay of leptin signal from this nucleus.

## PMV[PACAP] neurons are glutamatergic, monosynaptically connect to ARC and AVPV/PeN kisspeptin neurons, and PACAP directly stimulates activity in a regionally distinct subset of these neurons.

PMV[PACAP] neurons send projections to both regions of kisspeptin neuron populations, the AVPV/PeN and the ARC (*Figure 3a,b*, *Figure 3—figure supplement 1a,b*). These PMV[PACAP] neurons are glutamatergic (*Figure 3c*), though there are also PMV glutamatergic neurons that do not express PACAP. We used channelrhodopsin-assisted circuit mapping in order to determine if these close projections from glutamatergic, PACAPergic PMV neurons make direct monosynaptic contact with each subpopulation of kisspeptin neurons. In the ARC, where kisspeptin neurons participate in the regulation of GnRH pulsatility (*Clarkson et al., 2017*), and the expression of the PACAP receptor has been demonstrated (*Campbell et al., 2017*), blue light stimulation of PMV[PACAP] neurons at regular intervals (1 s) in brain slices led to immediate (<6 ms latency) excitatory postsynaptic currents in kisspeptin neurons. In female mice (n = 5), there was a monosynaptic glutamatergic connection between PMV[PACAP] neurons and ARC[kisspeptin] neurons (14/19 kisspeptin neurons tested throughout the ARC), and this was less frequently observed in non-kisspeptin ARC neurons (only 5/17 neurons received direct monosynaptic input from PMV[PACAP] neurons) (*Figure 3d*). Similarly, in the AVPV/PeN, where the kisspeptin neurons are thought to drive the large GnRH release that leads to the preovulatory LH surge (*Herbison, 2016*), the majority of AVPV/PeN[kisspeptin] neurons tested received direct input from PMV[PACAP] neurons (11/18 neurons connected) but only a small fraction of non-kisspeptin neurons assayed showed direct connectivity in response to activation of PMV[PACAP] neurons (2/13 neurons connected).

Because channelrhodopsin-assisted circuit mapping does not resolve the effects of slow neuropeptidergic neurotransmission, we isolated the effect of PACAP applied directly to kisspeptin neurons in both the AVPV/PeN and the ARC. We used brain slice calcium imaging to examine the responses of kisspeptin neurons located throughout both regions to PACAP (10 nM). Brain slices containing the AVPV and PeN or the rostral, middle and caudal aspects of the ARC were prepared from Kiss1-Cre;GCaMP6 mice (*Figure 3e*). PACAP was found to induce different effects on the basal and spontaneous calcium fluctuations observed in 34 AVPV/PeN[kisspeptin] neurons (four slices from each of 4 diestrous mice). Nine cells (25%) exhibited increases in the numbers of calcium fluctuations with, typically, no change in basal fluorescence, ten cells (29%) showed decreases in basal levels as well as the numbers of calcium fluctuations, and 15 neurons (46%) showed no response to PACAP application (*Figure 3f*). PACAP was found to induce region-dependent effects on fluorescence levels in ARC[kisspeptin] neurons. A total of 27 rostral, 64 middle, and 68 caudal neurons (each from 4 slices from four diestrous mice) were tested. While no cells in the rostral and middle ARC responded to PACAP, 17/68 (25%) of caudal kisspeptin neurons exhibited increases in fluorescence levels during PACAP application, with a decrease to basal fluorescence upon washout (*Figure 3h*). The remainder

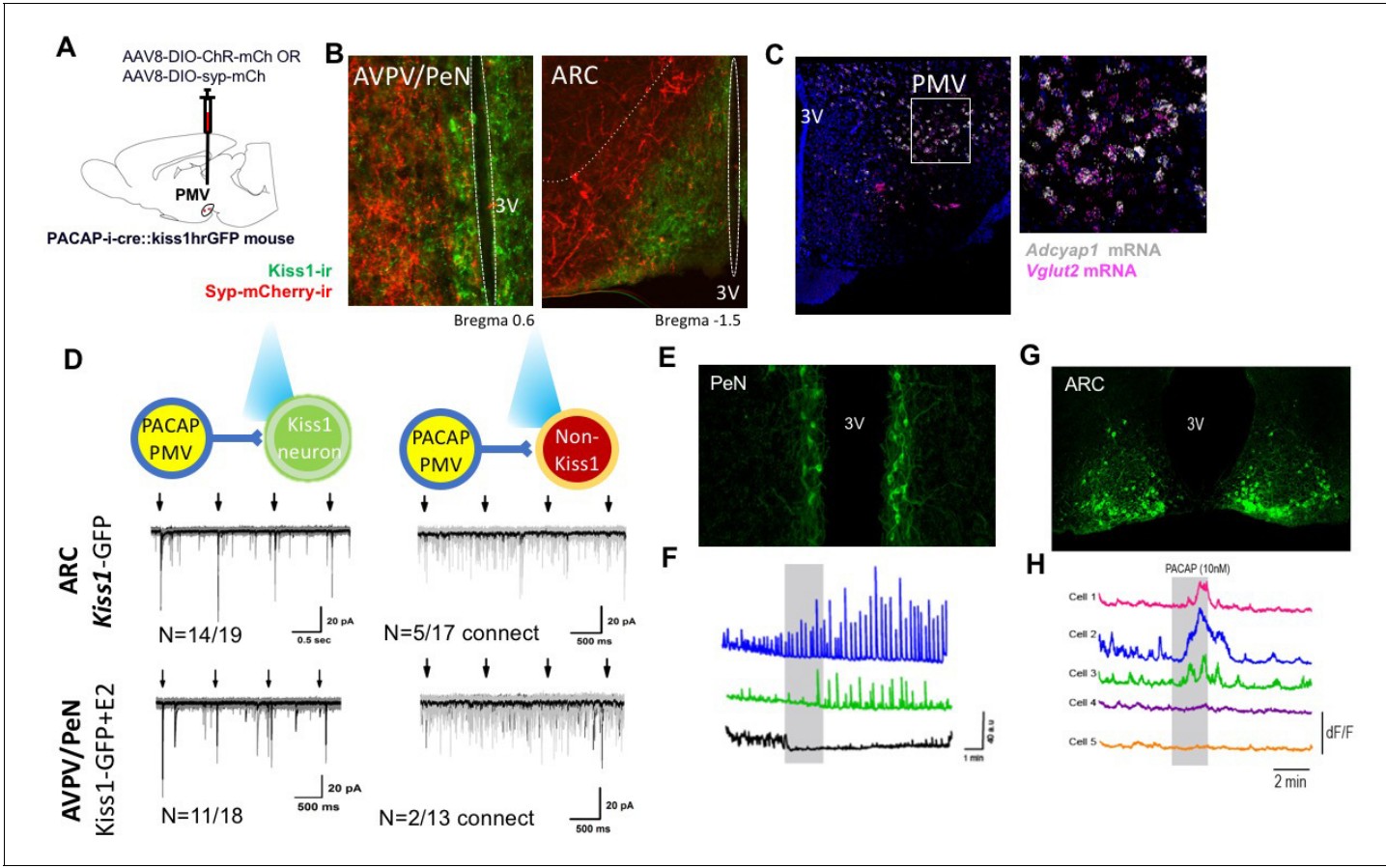

**Figure 3.** PMV^PACAP neurons monosynaptically contact a subset of ARC and AVPV Kisspeptin neurons. (**A**) Schematic representation of the site of injection of AAV8.2-eF1a-DIO-syp-mCherry-WPRE (projection tracing, n = 3) or AAV8-hSyn-DIO-ChR2(H134R)-mCherry (circuit mapping, n = 5). (**B**) Representative microphotographs depicting immunoreactivity of projections from PMV^PACAP neurons (red) and Kiss1 neurons (green) in the ARC and AVPV/PeN nuclei after injection of AAV8.2-eF1a-DIO-syp-mCherry-WPRE into the PMV in PACAP-i-cre female mice. (**C**) Representative double label ISH depicting co-localization of *Adcyap1 (white)* and *Vglut2 (pink)* mRNA in the PMV of female C57Bl/6 mice in a coronal section. Right image is an enlarged view of the boxed section in the left image. (**D**) Channelrhodopsin assisted circuit mapping (CRACM) analysis of projections from PACAP-i-cre mice injected with AAV-DIO-syp-ChR2-mcherry photo-stimulated in the ARC or AVPV of Kiss1-GFP mice. The majority of tested kisspeptin neurons in both regions receive direct excitatory input from PMV^PACAP neurons (14/19 in ARC, 11/18 in AVPV/PeN), and a small number of non-kisspeptin neurons in each region (5/17 and 2/13 respectively) do as well. Arrows depict blue light pulse (473 nm, 5 ms epoch) administered 1 s apart during the first 4 s of an 8 s sweep, repeated for a total of 30 sweeps per recorded cell. Black line shows the average of all sweeps, and gray lines are individual sweeps. (**E**) Representative photomicrograph of coronal section showing GCaMP6f fluorescence in the AVPV/PeN specific to kisspeptin neurons of a diestrous female Kiss1-GCaMP6f mouse. (**F**) Traces showing the effect of 10 nM PACAP (grey bar) on GCaMP6f fluorescence levels (delta F/F) in three kisspeptin cells recorded simultaneously from the AVPV/PeN indicating PACAP causes different responses in different populations of neurons in this region. (**G**) Representative photomicrograph of coronal section showing GCaMP6f fluorescence in the caudal arcuate nucleus specific to kisspeptin neurons of a diestrous female Kiss1-GCaMP6f mouse. (**H**) Traces showing the effect of 10 nM PACAP (grey bar) on GCaMP6f fluorescence levels (delta F/F) in five kisspeptin cells recorded simultaneously from the caudal ARC indicating PACAP causes increased activity in a subpopulation of kisspeptin neurons.
DOI: https://doi.org/10.7554/eLife.35960.008

The following figure supplement is available for figure 3:

**Figure supplement 1.** Representative images of PMV injection site for tracing and electrophyisiology experiments.
DOI: https://doi.org/10.7554/eLife.35960.009

showed no response to PACAP application. Together, with the channelrhodopsin circuit mapping experiments, these findings indicate that PMV^PACAP neurons project directly to kisspeptin neurons, but PACAP has direct stimulatory effect on a subpopulation of neurons located in the caudal aspect of the ARC, and interspersed through the AVPV/PeN. Furthermore, PACAP has a net inhibitory effect on another subpopulation of AVPV/PeN^kisspeptin neurons. Our experiments cannot distinguish if this PACAP activity originates from a particular population of PMV^PACAP neurons (e.g. LepR), yet

this is further evidence of the important role of the complex interactivity of kisspeptin neurons in generating both pulsatile and surge LH secretion (*Clarkson et al., 2017*).

## Discussion

In this study, we show that a large percentage of PMV neurons express PACAP (and glutamate), which modulates kisspeptin neurons, thus affecting fertility, estrous cycle, and puberty onset (*Figure 4*). The most striking finding is that deletion of PACAP from the PMV of adult female mice leads to >50% reduction in fecundity, a novel role for the neuromodulator specific to the PMV. Previous work has shown that PACAP is critical for fertility in whole body animal knockouts or in whole brain injection studies (*Jamen et al., 2000*; *Köves et al., 2003*; *Shintani et al., 2003*), but the mechanism and localization of action was unknown. This is the first study to show that a small population of hypothalamic glutamatergic LepR-expressing and PACAPergic neurons must express PACAP for normal timing of puberty onset and fertility in female mice.

PACAP effects have been shown to be sexually dimorphic, which is thought to be due to the estrogen response element of the PACAP receptor (*Ramikie and Ressler, 2016*). Interestingly, in this study, we observed that deletion of PACAP from the leptin-responsive neurons in female mice leads to delayed vaginal opening and delayed first estrus, both indirect markers of puberty onset, despite both groups having similar body weight. These data point to a crucial role of PACAP in the timing of sexual maturation in female mice, and aver the importance of future studies in males.

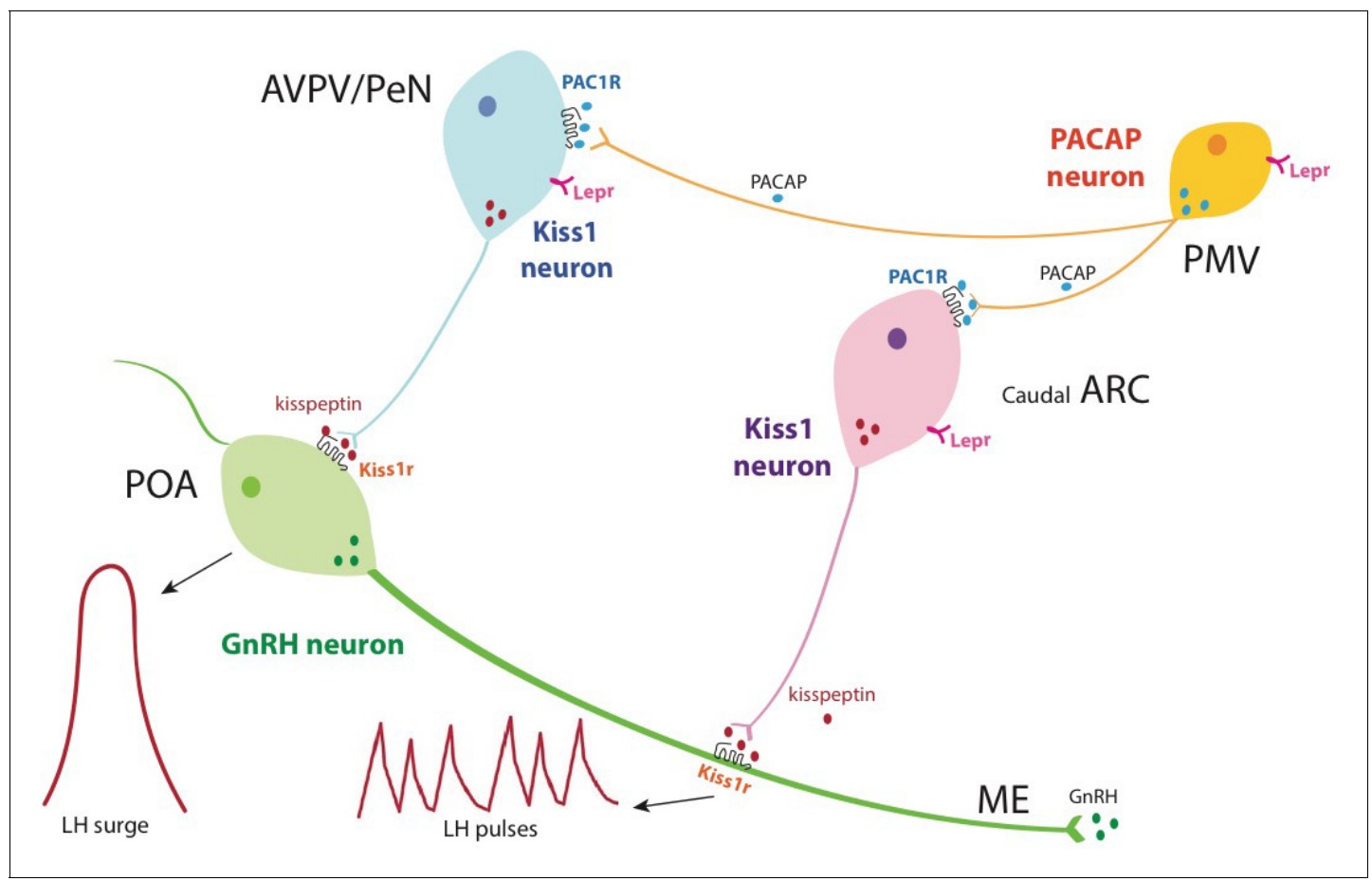

**Figure 4.** Schematic representation of the hierarchical inputs from PMV[PACAP] neurons to AVPV/PeN and ARC Kiss1 neurons. Briefly, PMV[PACAP] neurons regulate reproductive function by directly modulating subpopulations of AVPV/PeN[kisspeptin] and caudal ARC[kisspeptin] neurons, which leads to the regulation of the preovulatory GnRH/LH surge and pulsatile GnRH release, respectively, through the action of kisspeptin at the level of GnRH neurons. Through the direct action of leptin, PMV[PACAP] neurons mediate in part the metabolic regulation of reproductive function.
DOI: https://doi.org/10.7554/eLife.35960.010

Because PACAP and LepR are expressed in the gonads and other peripheral tissues, the potential deletion of PACAP from those areas could be possible in our LepR-i-cre; *Adcyap1*[fl/fl] mouse model. Use of the LepR-i-Cre mouse should mostly restrict deletion of PACAP to the brain, however, the long form of the leptin receptor may be expressed at low levels in the periphery (*Leshan et al., 2006*) and use of genetic cre-dependent manipulations can also lead to off-target deletion in neurons that express the LepR promoter during development but not in adulthood. Interestingly, in the gonads, LepR and PACAP are expressed in different cell types (*Duggal et al., 2002*; *Reglodi et al., 2012*; *Ryan et al., 2003*). There is a chance as well for off-target deletion of PACAP from cells (central or peripheral) that do not express LepR in adult animals. Still, both the LepR-i-cre mediated deletion of PACAP and the central AAV-delivery based manipulations recapitulate the effects of previously published ablation of PMV neurons (*Donato et al., 2011*). Taken together, our study demonstrates that the role of PACAP on the HPG axis is centrally mediated, specifically at the PMV.

As described above, PACAP seems to mediate the action of leptin (*Hawke et al., 2009*; *Tanida et al., 2013*; *Köves et al., 2003*). Circulating levels of leptin are essential for gonadotropin release (*Ahima et al., 1996*) and, as such, acute decreases in circulating leptin during fasting or undernourishment lead to hypogonadotropic hypogonadism. In this study, removal of PACAP from leptin-responsive neurons blunted the ability of exogenous leptin to increase LH levels after fasting, but direct deletion of PACAP from the PMV did not, suggesting that PACAP from the PMV is only partially necessary for the normal permissive role of leptin on the gonadotropic axis. We surmise that other factors (neuromodulators or neurons), or other leptin-responsive, PACAP-expressing regions of the brain, such as the VMH or supramammillary nucleus are likely involved in the transmission of the reproductive role of leptin (*Padilla et al., 2017*). Notably, PACAP neurons—and the PMV in general—are predominantly glutamatergic, yet our recent studies indicated that leptin's metabolic and reproductive actions are mediated by GABAergic transmission (*Martin et al., 2014*; *Vong et al., 2011*). While the absence of marked metabolic differences in these mice is in line with earlier studies (*Vong et al., 2011*), the hypogonadism observed in these mice are the first evidence of a divergent pathway of leptin to regulate metabolism and reproduction. Our data indicate that PACAP likely acts together with other neuromodulators produced in the same or separate LepR-expressing neurons through which leptin exerts its reproductive effect. Given that only a subset of kisspeptin neurons are directly activated by PACAP, and that a subpopulation of kisspeptin neurons is GABAergic, it is possible that these two populations of neurons (i.e. PMV[PACAP] and kisspeptin neurons) work in concert in response to different types of input from the PMV (i.e. PACAP-ergic or glutamatergic). Kiss1 and GnRH neuron firing patterns are very tightly regulated (*Herbison, 2016*) and further studies focusing on the neuromodulatory effects of PACAP (and other co-expressed peptides from the PMV[PACAP] neurons) will be required to disentangle the role of PMV neurons in the control of GnRH release.

We observed that the majority of AVPV/PeN[kisspeptin] and ARC[kisspeptin] neurons tested receive direct contact from PMV[PACAP] neurons, supporting a role for this neuropeptide in the control of the LH surge in females. Indeed, deletion of PACAP from the PMV either through viral delivery of cre recombinase to *Adcyap1*[fl/fl] mice or through crossing with LepR-i-cre mice led to a blunted LH surge and a smaller litter size, indicating an impairment in the ability of the mouse to ovulate and the presence of fewer mature follicles, which is also reinforced by fewer corpora lutea in the AAV-cre injected model. Moreover, the subfertility induced by both models of PACAP deletions led to irregular estrous cycles, indicating a role in LH pulsatility. Estrous cyclicity is determined by the ability of the mouse to mount an ovulatory process in the afternoon of proestrus and is the result of increasing levels of estradiol as a consequence of low frequency GnRH/LH pulses during diestrus (thought to be driven by ARC[kisspeptin]) (*Clarkson et al., 2017*), which drives more follicles to mature. The maturing follicles in turn synthesize more estradiol, which stimulates AVPV/PeN[kisspeptin] neurons (*Smith et al., 2005*; *Pinilla et al., 2012*), inducing an LH surge. These reproductive impairments suggest that PACAP from the PMV plays a role in both pulsatile and surge-like release of GnRH (*Figure 4*).

Within ARC[kisspeptin] neurons, the direct response to PACAP is specific to regional clusters of the caudal arcuate. These subpopulations of ARC[kisspeptin] cells have not been separately defined from the rest of the ARC population and this is the first time that PACAP activity has been shown to directly affect the activity of the ARC[kisspeptin] population, which express PACAP receptor (*Campbell et al., 2017*). ARC[kisspeptin] neurons are primarily involved in setting the tone for GnRH

pulsatility (*Clarkson et al., 2017*), a requisite for normal HPG axis function, therefore indicating a likely role of PMV$^{PACAP}$ in the fine tuning of kisspeptin and GnRH pulses. Similarly, the AVPV/PeN$^{kisspeptin}$ neurons are a heterogenous group based on their response to PACAP. This indicates that there are likely different receptors and second messenger systems activated by PACAP in subpopulations of AVPV/PeN$^{kisspeptin}$ neurons that may play different roles in the timing and magnitude of the preovulatory LH surge. As above, the ability of GnRH neurons to respond normally to kisspeptin in the absence of PMV$^{PACAP}$ signaling, strongly suggests an action on (as evidenced by *Figure 3*) or upstream of Kiss1 neurons. Furthermore, the compensatory LH rise in response to ovariectomy in these animals is also intact suggesting that PMV$^{PACAP}$ neurons are not required for sex steroid negative feedback, but instead may be involved in a separate pathway, such as the nutritional regulation of reproductive function. Together, these data indicate the neuromodulatory role of PACAP in the HPG axis is likely more subtle than the steroid hormone feedback, but remains important in its function.

In summary, we document the first evidence of a role for PACAP from the PMV in ovulatory cycling and subsequent fertility in females, which is required for normal reproduction activity (*Figure 4*). PACAP may also contribute to the reproductive role of leptin through the activation of glutamatergic, PACAPergic, leptin responsive neurons, but within the PMV, leptin signal transduction does not require PACAP expression. We also show for the first time that ARC$^{kisspeptin}$ and AVPV/PeN$^{kisspeptin}$ neurons are both heterogeneous populations, which can be defined by their response to PACAP, and this circuit is likely important for fine tuning of the HPG axis. These findings shed light on the complex mechanisms that underlie the neuroendocrine regulation of reproduction and may offer the platform to develop new treatments for reproductive disorders associated with metabolic impairments, such as anorexia nervosa, polycystic ovarian syndrome, or obesity.

## Materials and methods

### Animals and surgeries
#### Subjects
All animal care and experimental procedures were approved by the National Institute of Health, Beth Israel Deaconess Medical Center and Brigham and Women's Hospital Institutional Animal Care and Use Committee, protocol #05165. The Brigham and Women's Hospital is a registered research facility with the U.S. Department of Agriculture (#14–19), is accredited by the American Association for the Accreditation of Laboratory Animal Care, and meets the National Institutes of Health standards as set forth in the Guide for the Care and Use of Laboratory Animals (DHHS Publication No. (NIH) 85–23 Revised 1985). Mice were housed at 22–24°C with a 12 hr light (06:00)/dark (18:00) cycle with standard mouse chow (Teklad F6 Rodent Diet 8664) and water provided *ad libitum*. For behavioral studies, females between age 8 and 16 weeks were used. For electrophysiologic studies, female mice between age 5 and 12 weeks were used. All cre-driver and cre-reporter mice were used in the heterozygous state. All transgenic mice were maintained on a mixed background and have been described previously, except for the pacap-lox mice. Wild-type, *Adcyap1*$^{fl/fl}$ mice and PACAP-i-cre (*Krashes et al., 2014*) were maintained as separate litters and group housed according to sex. LepR-i-cre mice (*Leshan et al., 2006*) (Martin Myers, University of Michigan) were crossed to *Adcyap1*$^{fl/fl}$ and non-cre expressing homozygous flox allele offspring of the cross were used as littermate controls. Crosses were made between PACAP-i-cre and Kiss1hr-GFP (humanized *Renilla* Green Fluorescent Protein) (*Cravo et al., 2013*) (gift of Carol Elias, University of Michigan) for electrophysiology studies, and PACAP-i-cre or vglut2-cre and *Rosa26/CMV/Actin-loxSTOPlox-L10GFP* mice (L10-GFP, David Olson) for immunohistological detection.

To make the *Adcyap1*$^{fl/fl}$ mice, the lox-modified PACAP targeting construct was made by recombineering technology. To engineer the targeting vector, 5' homology arm, 3' homology arm and CKO region were amplified from mouse Sv129 BAC genomic DNA and confirmed by end sequencing (Cyagen biosciences, Santa Clara, CA). The two *loxP* sites flank the second exon and when recombined, create a frameshift mutation and truncated protein. The plasmid was electroporated into W4 ES cells and cells expanded from targeted ES clones were injected into C57Bl/6 blastocysts. Germline transmitting chimeric animals were obtained and then mated with flpE mice to delete the *frt*-site flanked neomycin selection cassette. The resulting heterozygous offspring were crossed to

generate homozygous *Adcyap1*<sup>fl/fl</sup> study subjects. All mice are thus on a mixed C57Bl/6J and 129Sv background. Offspring were genotyped by PCR using 2 primers (F: CCGATTGATTGACTACAGGC TCC and R: GTGTTAAACACCAGTTAGCCACGC) which detect the presence or absence of the 5' loxP site and a 3<sup>rd</sup> primer was used in conjunction with the forward primer (CKO-R GGGCTTTGATC TGGGAACTGAAG) to detect the recombination event. By generating mice homozygous for a germ-line cre-deleted allele (by cross to E2A-cre, Jax labs), we have established that the cre-deleted allele does not express intact *Adcyap1* mRNA (by PCR and by ISH, *Figure 2—figure supplement 1*).

## Subject history

For viral mediated PACAP deletion studies, *Adcyap1*<sup>fl/fl</sup> mice were randomly assigned to control or AAV-cre treated groups. For transgenic mice, groups with the presence or absence of Cre were determined by genotype of mouse and then randomly distributed. Animals were naïve to experimental testing before beginning the study. Multiple LH induction studies were conducted in the same animal subjects for each cohort, but each type of study was performed only once per cohort.

## Viral injections

Stereotaxic injections were performed as previously described in female mice between ages 6–8 weeks. Mice were anaesthetized with xylazine (5 mg/kg) and ketamine (75 mg/kg) diluted in saline (350 mg per kg) and placed into a stereotaxic apparatus (KOPF Model 963 or Stoelting). For postoperative care, mice were injected intraperitoneally with meloxicam (5 mg/kg). After exposing the skull via small incision, a small hole was drilled for injection. A pulled-glass pipette with a 20–40 nm tip diameter was inserted into the brain, and virus was injected by an air pressure system. A micromanipulator (Grass Technologies, Model S48 Stimulator) was used to control injection speed at 25 nl min−one and the pipette was withdrawn 5 min after injection. For electrophysiology experiments AAV8-hSyn-DIO-ChR2(H134R)-mCherry (University of North Carolina Vector Core; titer 1.3 × 1012 genome copies per ml) was injected unilaterally into the ventral premamillary region (PMV 25 nL, AP: −2.47, DV: −5.6, ML:+0.55) of PACAP-i-cre and Kiss1hr-GFP mice. For PACAP in vivo deletion studies, AAV8.CMV.HI.eGFP-Cre.WPRE.SV40 or AAV8.CMV.PI.eGFP.WPRE.bGH for control (University of Pennsylvania Vector Core; titer 1.01 × 10<sup>13</sup> and 1.39 × 10<sup>13</sup> respectively) was injected bilaterally into the PMV of *Adcyap1*<sup>fl/fl</sup> mice (same volume and coordinates). Bilateral hits were defined post-hoc by visual comparison with Allen Brain Atlas delineation of PMV region, using immunohistochemistry to check the spread of injection, and RNA ISH to confirm that *Adcyap1* was in fact deleted. Animals with only unilateral PMV *Adcyap1* deletion, or with deletion that extended to the VMH, were discarded from analysis.

For tracing studies from PACAP-ergic neurons, AAV8.2-eF1a-DIO-syp-mCherry-WPRE (Massachusetts Institute of Technology Virovek core; titer 2.23 × 10<sup>13</sup> genome copies per ml) was injected unilaterally to the PMV (same volume and coordinates). Mice were given a minimum of 2 weeks for recovery and 1 week for acclimation before being used in any experiments.

## Intracerebroventricular (ICV) injection

2–3 days before the experiment, the mice were briefly anesthetized with isoflurane and a small hole was bored in the skull 1 mm lateral and 0.5 mm posterior to bregma with a Hamilton syringe attached to a 27-gauge needle fitted with polyethylene tubing, leaving 3.5 mm of the needle tip exposed. Once the initial hole was made, all subsequent injections were made at the same site. For ICV injections, mice were anesthetized with isoflurane for a total of 2–3 min, during which time 5 μl of solution were slowly and continuously injected into the lateral ventricle. The needle remained inserted for approximately 60 s after the injection to minimize backflow up the needle track. Mice typically recovered from the anesthesia within 3 min after the injection. For hormonal analyses, blood samples (4 μl) were obtained from the tail before and 30 min after injection. Recombinant mouse Leptin (2 ug/5 ul) was purchased from Preprotech. Mouse kisspeptin-10 (Kp-10) (1 nmol/5 ul) was obtained from Phoenix Pharmaceuticals. The drugs were dissolved in saline (0.9% NaCl). The dose and time of collection were selected based on our previous studies (*Navarro et al., 2015*).

## Estrous cycle monitoring

Female mice are spontaneous ovulators and typically have a 4 day estrous cycle that consists of four stages: proestrus, estrus, metestrus/diestrus I and diestrus II. Mice were habituated to handling and vaginal smears before behavioral experiments. Mice were swabbed daily at 9:00am to check for consistent cycling (*Caligioni, 2009*) and underwent blood sampling at 10:00am, except for LH surge, which occurred just after the onset of the dark cycle. Using a disposable pipette, a small volume of physiological sterile water was placed near the opening of the vagina and 10 ul was flushed in three repetitions with minimal insertion to avoid pseudopregnancy. The smear was displaced onto a glass slide and after air-drying, was examined under brightfield microscope at 20X. The identification of estrous stages was based on characteristic cell type appearance observed and the density of each cell type in the vaginal secretion (*Caligioni, 2009*). Estrous stage was identified by changes in cell appearance across the cycle that reflects circulating gonadal steroids. A cycle was considered as the time between two consecutive estrous phases with at least one day of diestrous phase in between them.

## Fecundity

Female study mice were paired with 8 week old C57Bl/6 male mice for 1 week, at which time the males were removed from the cages and females were left individually housed with cage enrichment, and monitored for signs of pregnancy. Post-birth, females were paired again for 1 week with a new C57Bl/6 male to ensure any potential infertility was due to the female only.

## Tail-tip bleeding

Blood sampling was performed after a single excision of the tip of the tail. Tail was cleaned with saline and then massaged to take a 4 ul blood sample from the tail tip with a pipette. Whole blood was immediately diluted in 116 ul of 0.05% PBST, vortexed, and frozen on dry ice. Samples were stored at −80°C for a subsequent LH ELISA (*Steyn et al., 2013*).

## Ovariectomies

The time-course of changes in the circulating levels of LH in response to ovarectomy (OVX) was explored in female mice to assay for feedback mechanisms of gonadal steroids on HPG axis. Groups of adult female mice were subjected to bilateral OVX via abdominal route, as previously described (*García-Galiano et al., 2012*). Blood samples were obtained before and 7-d after OVX to use for LH ELISA (*García-Galiano et al., 2012*).

## Induction of LH surge/E2 replacement

Immediately after OVX, capsules containing 0.625 ug of 17-$\beta$ estradiol dissolved in sesame seed oil were implanted subcutaneously via a small midscapular incision at the base of the neck. Silastic tubing (15 mm long, 0.078 in inner diameter, 0.125 in outer diameter; Dow Corning) was used for capsule preparation. Whole blood samples were collected for LH analysis 2 days after surgery in the morning (10:00am EST) and in the evening just after lights off (19:00) (*Dror et al., 2013*).

## Electrophysiology

Female animals in diestrous stage were deeply anesthetized and decapitated. Brains were quickly removed into ice-cold cutting solution consisting of (in mM) 72 sucrose, 83 NaCl, 2.5 KCl, 1 NaH2PO4, 26 NaHCO3, 22 glucose, 5 MgCl2, 1 CaCl2, oxygenated with 95% O2/5% CO2, measured osmolarity 310–320 mOsm/l. 300-m-thick coronal sections were cut with a Leica VT1000S vibratome and incubated in oxygenated cutting solution at 34°C for 45 min. Slices were transferred to oxygenated aCSF (126 mM NaCl, 21.4 mM NaHCO3, 2.5 mM KCl, 1.2 mM NaH2PO4, 1.2 mM MgCl2, 2.4 mM CaCl2, 10 mM glucose) and stored in the same solution at room temperature (20–24°C) for at least 60 min before recording. A single slice was placed in the recording chamber where it was continuously superfused at a rate of 3–4 ml per min with oxygenated aCSF. Neurons were visualized with an upright microscope (SliceScope, Scientifica) equipped with infrared differential interference contrast and fluorescence optics. Borosilicate glass microelectrodes (5–7 M) were filled with internal solution.

For CRACM experiments, recordings were obtained using a Cs+-based low-Cl− internal solution consisting of (in mM) 135 cesium methanesulfonate, 10 HEPES, 1 EGTA, 4 MgCl2, 4 Na2-ATP, 0.4 Na2-GTP, 10 disodium phosphocreatine (pH 7.3 adjusted with CsOH; 295 mOsm kg−1; $E$Cl = −70 mV). Initially, light- evoked EPSCs or IPSCSs were recorded in whole-cell voltage-clamp mode, with membrane potential clamped at $V$h = −70 mV or 0 mV, respectively. Once the excitatory nature of the recorded inputs was established, picrotoxin was included to isolate glutamatergic currents. The light-evoked EPSC protocol consisted of four blue light pulses (473 nm wavelength, 5 ms) administered 1 s apart during the first 4 s of an 8 s sweep, repeated for a total of 30 sweeps. Evoked EPSCs with short latency (<6 ms) upon light stimulation were considered to be light-driven. As discussed by others, such currents are most likely monosynaptic (*Petreanu et al., 2007*). Number of animals used for CRACM experiments: PMV^PACAP→ ARC^kisspeptin, n = 5; PMV^PACAP→ AVPV/PeN^kisspeptin, n = 5. The PMV^PACAP neurons were labeled with AAV8-DIO-ChR2-mCherry injected unilaterally into the PMV of *PACAP-i-cre;kiss1hr-GFP* mice. To visualize kisspeptin neurons in the AVPV/PeN, estrogen was injected subcutaneously 2 days prior to experiments where AVPV/PeN neurons were targeted for recording (this was not required for visualizing the ARC kiss1 population). For consistency, we recorded from both regions in the setting of estrogen treatment, and there was no difference in the connections between neurons when comparing connections in animals that received estrogen and those that did not.

## Calcium imaging

Mice were generated by crossing Kiss*1*-Cre^+/- (Kiss1^tm2(Cre-GFP)Coll), (*Yeo et al., 2016*) generated in the Herbison lab (University of Otago, NZ), and homozygous floxed GCaMP6f (Ai95(RCL-GCaMP6f)-D, also known as 129S-*Gt(ROSA)26Sor^tm95.1(CAG-GCaMP6f)Hze*) (*Madisen et al., 2015*) lines to generate mixed background 129S6Sv/Ev C57Bl/6 Kiss1-*Cre;*GCaMP6f-*lox-STOP-lox* (Kiss1-GCaMP) mice. Mice (n = 4 per region of interest) were housed under a 12:12 hr lighting schedule (lights on at 6:00 A.M.) with *ad libitum* access to food and water and females investigated at the diestrous phase of the cycle. These experiments were approved by the University of Otago Animal Welfare and Ethics Committee.

Calcium imaging of kisspeptin neurons was undertaken in acute brain slices using previously published methodology (*Piet et al., 2015*). In brief, 300 μm-thick brain slices containing the AVPV and PeN rostral, middle and caudal regions of the ARN were prepared and constantly perfused (1 mL/min) with 30°C, 95%$O_2$/5%$CO_2$ equilibrated, artificial cerebrospinal fluid (aCSF) comprised of (mM) NaCl 120, KCl 3, NaHCO3 26, NaH2PO4 1, CaCl2 2.5, MgCl2 1.2 and glucose 10, and containing 0.5 μM TTX, 100 μM picrotoxin, 10 μM CNQX, and 40 μM AP5 to block all synaptic transmission. Slices were placed under an upright Olympus BX51W1 microscope and multiple individual cells in a plane of focus visualized through a 40x immersion objective using a xenon arc light source (300 W; filtered by a GFP filter cube (excitation 470–490 nm, Chroma)) and a DG-4 shutter (Sutter Instruments). Epi-fluorescence (495 nm long pass and emission 500–520 nm) was collected at 2 Hz using a Hamamatsu ORCA-ER digital CCD camera. The effects of PACAP on kisspeptin neuron fluorescence were assessed by performing basal image acquisition (4 min) and adding 10 nM PACAP to the aCSF for a two-min period before switching back to aCSF only for a further 4 min wash period. Regions of interest over individual, non-overlapping, and in-focus fluorescent somata were selected and analyzed using ImageJ software and custom R scripts (*Clarkson et al., 2017*). Individual cells were considered to have responded if they exhibited an increase in fluorescence that was greater than their mean basal level +2 standard deviations (4 min prior to drug). For AVPV/PeN^kisspeptin neurons, a 25% change in the frequency of individual calcium events between the basal and test +wash periods was considered a response. For visualizing data, values are presented as relative fluorescence changes using $\frac{\Delta F}{F} = \frac{Ft-F}{F} *100$ where $F$ is the baseline fluorescence intensity calculated as the mean fluorescence intensity over the 4 min period preceding drug applications and $F_t$ is the fluorescence measured.

## Fluorescence in situ hybridization

Brains were sectioned coronally at 12 μm using a cryostat, thaw- mounted onto electrostatically clean slides, and stored at 80°C until postfixed. Prior to hybridization, sections were postfixed in 4% paraformaldehyde, rinsed in 0.1 M PBS (pH 7.4), equilibrated in 0.1 M triethanolamine (pH 8.0), and

acetylated in triethanolamine containing 0.25% acetic anhydride. Standard in vitro transcription methods were used to generate both sense and antisense riboprobes recognizing PACAP transcript, which were subsequently diluted in hybridization cocktail (Amresco) with tRNA. Sections were hybridized overnight at 60°C with either digoxigenin (DIG)-labeled riboprobes. After hybridization, slides were treated with RNase A and stringently washed in 0.5x SSC at 56°C for 30 min. Slides were then incubated with an antibody against DIG conjugated to horseradish peroxidase (HRP; Roche) overnight at 4°C. Riboprobe signal was further enhanced using the TSA-Plus fluorophore system with Alexa Fluor conjugated streptavidin (LifeTechnologies). Image capture was performed using fluorescent microscopy (Olympus VS120 slide scanner microscope) and analyzed with image J software. The primer sequences for the riboprobe generation are 5'-ccaatgaccatgtgtagcg and 5'- atcagaccagaagacgaggc. For dual fluorescence ISH, we used probes for *Adcyap1* and *Vglut2* obtained from ACDBio and used the RNAscope method per their protocol (ACDBio).

## Brain tissue preparation for immunohistochemistry

Animals were terminally anesthetized with 7% chloral hydrate diluted in saline (350 mg/kg) and transcardially perfused with phosphate-buffered saline (PBS) followed by 10% neutral buffered formalin (PFA). Brains were removed, stored in the same fixative for 2 hr, transferred into 20% sucrose at 4°C overnight, and cut into 30 mm sections on a freezing microtome (Leica) coronally into four equal series. A single series of sections per animal was used in the histological studies.

## Immunohistochemistry

Brain sections were washed in PBS with Tween-20, pH 7.4 (PBST) and blocked in 3% normal donkey serum/PBST for 1 hr at room temperature. Then, brain sections were incubated overnight at room temperature in blocking solution containing primary antiserum (rat anti-mCherry, Life Technologies, 1:1,000; chicken anti- GFP, Life Technologies, 1:1000 (Millipore, #AB9754). The next morning sections were extensively washed in PBS and then incubated in Alexa- fluorophore secondary antibody (Molecular Probes, 1:1,000) for 1 hr at room temperature. After several washes in PBS, sections were mounted on gelatin-coated slides and fluorescence images were captured with Olympus VS120 slide scanner microscope.

For pStat3 immunohistochemistry studies, Mice were injected with 5 mg/kg recombinant leptin 2 hr before perfusion (as above). Brain sections were washed in 0.1 M phosphate-buffered saline, pH 7.4, followed by incubation in 5% NaOH and 0.3% $H_2O_2$ for 2 min, then with 0.3% glycine (10 min), and finally with 0.03% SDS (10 min), all made up in PBS. Sections were blocked in 3% normal donkey serum/0.25% Triton X-100 in PBS for 1 hr at room temperature and then incubated overnight at room temperature in blocking solution containing 1/250 rabbit anti-pStat3 (Cell Signaling, #9145) and 1/1,000 chicken anti-GFP (Life Technologies, #A10262). The next morning sections were extensively washed in PBS and then incubated in 1/250 donkey anti-rabbit 594 (Molecular Probes, R37119) and 1/1,000 donkey anti-chicken 488 (Jackson ImmunoResearch, 703-545-155) for 2 hr at room temperature. After several washes in PBS, sections were mounted onto gelatin-coated slides and fluorescent images were captured with Olympus VS120 slide scanner microscope.

## Brain punches

Brains were rapidly extracted, cooled in ice-cold DMEM/F12 medium for 5 min and then placed, ventral surface up, into a chilled stainless steel brain matrix (catalog no. SA-2165, Roboz Surgical Instrument Co., Gaithersburg, MD). Using anatomical markers, brains were blocked to obtain a single coronal section containing the entire PMV,~2 mm thick. The PMV was microdissected by knife cuts at its visually approximated dorsolateral borders under 5x magnification.

## RNA Expression (qRT-PCR)

Total RNA from PMV was extracted from frozen tissue collected using TRIzol reagent (Invitrogen) followed by chloroform/isopropanol extraction. RNA was quantified using a NanoDrop 2000 spectrophotometer (Thermo Scientific) and 1 μg of RNA was reverse transcribed using Superscript III cDNA synthesis kit (Invitrogen). Quantitative real-time PCR assays were performed on an ABI Prism 7000 sequence detection system, and analyzed using ABI Prism 7000 SDS software (Applied Biosystems). The cycling conditions were as follows: 2 min incubation at 95°C (hot start), 45 amplification cycles

(95°C for 30 s, 60°C for 30 s, and 45 s at 75°C, with fluorescence detection at the end of each cycle), followed by melting curve of the amplified products obtained by ramped increase of the temperature from 55°C to 95°C to confirm the presence of single amplification product per reaction. *Adcyap1* expression was detected using primers PACAP F- GAA ACC CGC TGC AAG ACTT/PACAP R – CGA CAT CTC TCC TGT CCGC and was normalized with housekeeping gene *Rpl19*.

## Statistical analysis

All data are expressed as the mean ± SEM for each group. A two tailed unpaired t-Student test or a one- or two-way ANOVA test followed by Bonferroni or Fisher's post-hoc test was used to assess variation among experimental groups. Significance level was set at $p < 0.05$. All analyses were performed with GraphPad Prism Software, Inc (San Diego, CA).

## Acknowledgements

This work was supported by NIH R01 HD019938 and R01 HD082314 to UBK; R00 HD071970 and R01 HD090151 to VMN; NHLBI 5T32HL007374-36, NCATS UL1 TR001102 and Harvard Medical School Dupont Warren Fellowship to RAR; R01 DK075632, R01 DK096010, R01 DK089044, R01 DK111401, P30 DK046200, P30 DK057521 to BBL.

## Additional information

### Funding

| Funder | Grant reference number | Author |
|---|---|---|
| National Heart, Lung, and Blood Institute | 5T32HL007374-36 | Rachel A Ross |
| National Center for Advancing Translational Sciences | UL1 TR001102 | Rachel A Ross |
| Harvard Medical School | Dupont Warren Fellowship | Rachel A Ross |
| National Institutes of Health | R01 HD082314 | Ursula B Kaiser |
| National Institutes of Health | R01 HD019938 | Ursula B Kaiser |
| National Institutes of Health | R01 DK075632 | Bradford B Lowell |
| National Institutes of Health | R01 DK089044 | Bradford B Lowell |
| National Institutes of Health | R01 DK111401 | Bradford B Lowell |
| National Institutes of Health | P30 DK046200 | Bradford B Lowell |
| National Institutes of Health | P30 DK057521 | Bradford B Lowell |
| National Institutes of Health | R01 DK096010 | Bradford B Lowell |
| National Institutes of Health | R01 HD090151-A1 | Victor M Navarro |
| National Institutes of Health | R00 HD071970 | Victor M Navarro |

The funders had no role in study design, data collection and interpretation, or the decision to submit the work for publication.

### Author contributions

Rachel A Ross, Conceptualization, Resources, Data curation, Formal analysis, Validation, Investigation, Visualization, Methodology, Writing—original draft, Project administration, Writing—review and editing; Silvia Leon, Resources, Data curation, Formal analysis, Investigation, Methodology; Joseph C Madara, Danielle Schafer, Anne MJ Verstegen, Resources, Methodology; Chrysanthi Fergani, Resources, Formal analysis, Investigation, Methodology; Caroline A Maguire, Investigation, Methodology; Emily Brengle, Data curation, Methodology; Dong Kong, Resources; Allan E Herbison, Bradford B Lowell, Conceptualization, Resources, Methodology; Ursula B Kaiser, Conceptualization, Supervision, Visualization; Victor M Navarro, Conceptualization, Resources, Data curation, Software,

Formal analysis, Supervision, Funding acquisition, Validation, Investigation, Visualization, Methodology, Writing—original draft, Project administration, Writing—review and editing

### Author ORCIDs
Allan E Herbison [ID] http://orcid.org/0000-0002-9615-3022
Victor M Navarro [ID] http://orcid.org/0000-0002-5799-219X

### Ethics

Animal experimentation: All animal care and experimental procedures were approved by the National Institute of Health, Beth Israel Deaconess Medical Center and Brigham and Women's Hospital Institutional Animal Care and Use Committee. protocol #05165.

### Decision letter and Author response

Decision letter https://doi.org/10.7554/eLife.35960.013
Author response https://doi.org/10.7554/eLife.35960.014

# Additional files

### Supplementary files

• Transparent reporting form
DOI: https://doi.org/10.7554/eLife.35960.011

### Data availability

All data generated or analysed during this study are included in the manuscript and supporting files.

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
