## [Decision Letter]

Thank you for submitting your article "PACAP neurons in the ventral premammillary nucleus regulate reproductive function in the female mouse" for consideration by *eLife*. Your article has been reviewed by three peer reviewers, one of whom is a member of our Board of Reviewing Editors, and the evaluation has been overseen by a Senior Editor. The reviewers have opted to remain anonymous.

The reviewers have discussed the reviews with one another and the Reviewing Editor has drafted this decision to help you prepare a revised submission.

Summary:

Ross and colleagues investigated the role of PACAP expressing neurons in the ventral premamillary nucleus (PMV) in regulating parameters of reproductive function. They used a combination of genetic, optogenetic, and physiological approaches to demonstrate that PACAP neurons in the PMV express leptin receptors and innervate Kisspeptin neurons to regulate reproduction. Deletion of PACAP from the PMV, either by crossing with leptin receptor-Cre mice, or through viral-mediated expression of Cre in the PMV, caused changes in gonadotropin secretion and reduced fecundity. Deleting PACAP in the PMV delays puberty onset and reproductive function in females.

Electrophysiological studies and calcium imaging confirmed alterations of postsynaptic kisspeptin neurons that likely contribute to the observed reproductive dysfunction. The authors propose a sex specific role of PACAP neurons in the PMV to relay nutritional state to regulate GnRH release by modulating the activity of kisspepting neurons, thereby regulating reproduction. Collectively, the studies are well described, provide novel data and are of potential interest. However, several issues need to be addressed.

Essential revisions:

The authors propose a sex specific role of PACAP and allude to neural circuit differences, but only show results from females. The authors show that in female mice there is connectivity between PMV PACAP neurons and ARC Kisspeptin neurons. Is this the same in males? And could that connectivity difference explain the lack of effect on male fertility (this sexually dimorphic response is very interesting, and explaining the circuitry that underlies it would be phenomenal)? Likewise, is there different PACAP or PACAP-Leptin expression in M v F mice? If male data is not formally included, the sex difference assertion should be removed.

More details regarding the anatomic experiments are needed. Were the AAV injections unilateral or bilateral? If the latter, how were "hits" defined? In addition, more photomicrographs and/or drawings would help in assessment of the degree of deletion and/or placement of the channel rhodopsin.

Please define in more detail the control groups used in the PACAP deletion studies. Similarly, what was the control virus expressing? What age was this done? Since the lepR deletion suggests a developmental deficit, how does one reconcile these findings?

The data in this manuscript add to the expanding literature identifying circuits that convey metabolic signals to hypothalamic regions that regulate various aspects of reproduction. The array of technical approaches is impressive and the direct interrogation of cellular relationships and neuronal signaling has provided data that supports the authors' conclusions. The results are clearly illustrated, and the supplementary data is warranted. The channelrhodopsin-assisted circuit mapping studies provide strong evidence of direct projects from PACAP neurons to kisspeptin neurons. The quantitative comparisons in relative frequency of inputs to kisspeptin neurons in the ARH and AVPV/PeN are welcome, but the sampling does not seem sufficient to support some of the statements. For example, recording from 13 neurons is insufficient to state that, "a small fraction of non-kisspeptin neurons showed direct connectivity…" or that recording from 18 neurons supports the statement that, "the majority of AVPV/PeN kisspeptin neurons received direct input from PMV PACAP neurons…" (Discussion section). It is also surprising that calcium imaging was not used to evaluate responses to PACAP in the AVPV/PeN, as it was in the ARC.

The methods are clearly described, and sufficient detail is provided. However, the figure legends are wholly inadequate. The reader should have a much better idea what they are looking at without having to combine comments in the text with sections of the methods. What is listed here is more of a series of titles for figures, rather than an explanation of what is depicted. This is likely a minor oversight because these authors have included informative figure legends elsewhere.

---

## [Author Response]

Essential revisions:The authors propose a sex specific role of PACAP and allude to neural circuit differences, but only show results from females. The authors show that in female mice there is connectivity between PMV PACAP neurons and ARC Kisspeptin neurons. Is this the same in males? And could that connectivity difference explain the lack of effect on male fertility (this sexually dimorphic response is very interesting, and explaining the circuitry that underlies it would be phenomenal)? Likewise, is there different PACAP or PACAP-Leptin expression in M v F mice? If male data is not formally included, the sex difference assertion should be removed.

We have removed the references to male mice, as to complete the set of data would take longer than allotted for this revision. We agree that the sexually dimorphic behavior response is very interesting, and worthy of more complete study, and will consider it for a future submission.

More details regarding the anatomic experiments are needed. Were the AAV injections unilateral or bilateral? If the latter, how were "hits" defined? In addition, more photomicrographs and/or drawings would help in assessment of the degree of deletion and/or placement of the channel rhodopsin.

Thank you for this request for clarification. This information is included in the Materials and methods section. We have added a line to clarify how hits were defined as requested in this section.

We have included a schematic of the viral spread around the PMV for the AAV-cre injections in Figure 2A (and there is an example of deletion in Figure 2B by ISH). We have now added a photomicrograph example of viral placement for the ChR and synaptophysin studies as Figure 3—figure supplement 1 in addition to the schematic, which was intended to represent both ChR and synaptophysin studies.

Please define in more detail the control groups used in the PACAP deletion studies. Similarly, what was the control virus expressing? What age was this done? Since the lepR deletion suggests a developmental deficit, how does one reconcile these findings?

Materials and methods section: Behavioral studies used females age 8-16 wks, details the control virus and tells what the control animals were. We now state the age range for surgical injections in this section as well.

We hope that these additions and the improved descriptions in the figure legends improve the clarity and detail of the experimental methods.

Because we wanted to delete PACAP in a regionally specific way, we are unable to perform AAV-cre based deletion of PACAP from the PMV to compare to the developmental findings of the LepR-cre mediated conditional KO. We have added a statement about this (relating also to the comment below regarding developmental off-target effects of cre deletion in the genetic KO) to the Discussion section.

The data in this manuscript add to the expanding literature identifying circuits that convey metabolic signals to hypothalamic regions that regulate various aspects of reproduction. The array of technical approaches is impressive and the direct interrogation of cellular relationships and neuronal signaling has provided data that supports the authors' conclusions. The results are clearly illustrated, and the supplementary data is warranted. The channelrhodopsin-assisted circuit mapping studies provide strong evidence of direct projects from PACAP neurons to kisspeptin neurons. The quantitative comparisons in relative frequency of inputs to kisspeptin neurons in the ARH and AVPV/PeN are welcome, but the sampling does not seem sufficient to support some of the statements. For example, recording from 13 neurons is insufficient to state that, "a small fraction of non-kisspeptin neurons showed direct connectivity…" or that recording from 18 neurons supports the statement that, "the majority of AVPV/PeN kisspeptin neurons received direct input from PMV PACAP neurons…" (Discussion section). It is also surprising that calcium imaging was not used to evaluate responses to PACAP in the AVPV/PeN, as it was in the ARC.

Thank you for these comments. We have toned down the quantitative comparisons to indicate that these refer only to the subset of neurons from which we recorded. We have also added calcium imaging data for the AVPV/PeN response to PACAP.

The methods are clearly described, and sufficient detail is provided. However, the figure legends are wholly inadequate. The reader should have a much better idea what they are looking at without having to combine comments in the text with sections of the methods. What is listed here is more of a series of titles for figures, rather than an explanation of what is depicted. This is likely a minor oversight because these authors have included informative figure legends elsewhere.

Thank you for pointing out this oversight. We have included more information in the figure legends as requested, and hope that this improves the clarity and readability of the figures.